# LANGUAGE AS KERNELS

## ABSTRACT

In the realm of natural language understanding, the synergy between large language models (LLMs) and prompt engineering has unfurled an impressive tapestry of performance. Nonetheless, this prowess has often been overshadowed by the formidable computational resource requirements, rendering LLMs inaccessible in resource-constrained milieus. In this study, we embark on a journey to reconcile this paradox by introducing a nimble and elegant solution — the kernel machine paradigm. Within these hallowed pages, we present a compelling proof, demonstrating the mathematical equivalence of zero-shot learning and kernel machines. This novel approach, marked by its computational thriftiness, bestows upon us the ability to harness the latent potential of LLMs, even when confined to the humble CPUs. The marriage of this approach with neural nets, renowned for their boundless abstraction capabilities, culminates in remarkable accomplishments with in the realm of language understanding. Our paramount contribution lies in unveiling a path less traveled, where the integration of kernel machines and LLMs unveils a promising vista, enabling the realization of sophisticated language processing tasks in resource-constrained environments.

## 1 INTRODUCTION

Language, as the bedrock of human communication, has intrigued and stimulated explorations for centuries. The advent of deep learning has introduced a new dimension to this exploration. The quest to understand and replicate the nuances of language has become a cornerstone of contemporary AI research (Bengio et al., 2013). Large Language Models (LLMs) (Vaswani et al., 2017; Devlin et al., 2018; Brown et al., 2020) have surfaced as potent tools in this quest, demonstrating impressive performance in understanding not only the cultural evolution of semantics, such as entailments and sentiments, but also grammatical transitions (Wang et al., 2019a). As a rich prior, these models have ushered in the era of *zero-shot learning* (Larochelle et al., 2008; Wang et al., 2019b; Wei et al., 2021). In this era, models can adapt to unseen domains despite a limited training corpus.

However, the computational resource requirements for fine-tuning such models with large parameters for zero-shot learning have been a source of concern, rendering them inaccessible in distributed environments such as connected devices or third-party vendors, where each device relatively accommodates less resources (Raina et al., 2009; Vanhoucke et al., 2011; Austerweil & Zoran, 2019). In this paper, we propose a novel solution to this problem by leveraging the power of the LLMs with kernel machines (Cortes & Vapnik, 1995), a class of algorithms known for their computational efficiency and versatility. We introduce a new approach, Support Vector Generation (SVG), that combines the generative capabilities of the pre-trained rich sequential models online such as LLMs with the computational thriftiness of kernel machines, which allows us to obtain the training data for zero-shot learning tasks without the need for high-performance computing.

In the next section, we provide an overview of the two independent branches of kernel machines and language models, and in Section 3, we merge the two branches to open up the kernel machine paradigm. In Section 4, we propose our framework, SVG, which formulates a rich decision boundary for classification without *any* training samples. In Section 5, we demonstrate the computational efficiency of SVG with General Language Understanding Evaluation (GLUE) benchmark (Wang et al., 2019a) and discuss various applications in Section 6 to conclude our paper.

## 2 RELATED WORK

Before the deep learning era, kernels (*i.e.* "similarity") were widely employed to measure the distance between a pair of inputs, especially sequences of information such as web documents or protein sequences, which can be of variable length. The representative one is the cosine similarity between TF-IDF vectors, which gives good results for information retrieval (Manning et al., 2008), with a probabilistic interpretation given in (Elkan, 2005). String kernels (Rasmussen & Williams, 2006; Hastie et al., 2009), which compare the number of substrings in common, have been used for amino acid sequences. The string kernel is known as a scalable Mercer's kernel, which can be computed in linear time for a length of string using suffix trees (Leslie et al., 2003; Vishwanathan et al., 2003; Shawe-Taylor & Cristianini, 2004). One of the string kernels which consider strings of a fixed length $k$ is known as the $k$-spectrum kernel, and has been used to classify proteins into SCOP superfamilies (Leslie et al., 2003). The string kernel has been generalized to compare trees (Collins, 2002), which is useful for parse trees and evolutionary trees. In computer vision, a pyramid match kernel (Grauman & Darrell, 2007) is used to compare two images of feature vectors obtained from SIFT (Lowe, 1999). One of the drawbacks of these kernels is that they cannot capture the recurring structure of strings, because they merely assume the document as a bag-of-words (or bag-of-features), and only care about the frequency of the words.

The paradox between the number of parameters of a neural network and its generalisation performance has been one of the unsolved problems in the field of deep learning: traditional learning theories state that if the parameters of a model exceed the number of samples, the model is overtrained and cannot adapt sufficiently to unknown samples. One hypothesis for this is that neural networks acquire the ability to "interpolate" between any two samples after having fully memorised all samples, which also has experimental evidence in the form of the double descent phenomena (Nakkiran et al., 2021). Nevertheless, another problem with this paradox is the derivation of explicit optimal solutions, *i.e.* how to regularise a model with degrees of freedom that go beyond the train data. Zhang et al. (2016) map the MNIST samples into a dual space and optimise the hinge loss with a "kernel trick" to obtain a regularised explicit representation consisting of the weights of each sample and the Gram matrix, which is a similarity representation between each sample. This solution also showed higher performance than the expected value of the solution optimised simply by the stochastic gradient descent, suggesting a strong relationship between neural networks and kernels.

Self-attention (Vaswani et al., 2017) is a powerful building block that has enabled the development of large language models (LLMs) such as BERTs and GPTs, which can quickly capture the in-context relationships between tokens in a text sample. LLMs have been used for various tasks including text generation, question answering, and dialogue generation (Radford et al., 2019; Brown et al., 2020; Radford et al., 2018; Zhang et al., 2019). Recent research has focused on controlled text generation, generating text that adheres to a set of constraints while being fluent and relevant to the given context (Zhu et al., 2019; Liu et al., 2019). Zero-shot learning, a more challenging task, uses transfer learning or data augmentation to overcome data scarcity and computational constraints (Gao et al., 2020; Meng et al., 2022). Despite its computational advantage, self-attention requires the use of GPUs to accommodate the large number of parameters ($|\theta| \sim 10^{10}$) needed to memorize all examples. This can be computationally expensive and may not be feasible for certain applications.

## 3 KERNEL MACHINES ARE ZERO-SHOT LEARNERS

Suppose we are a data scientist in a company who has been assigned a task of sentiment analysis, checking whether the given text represents a negative or positive review for each film manufactured by the client. In a zero-shot learning scenario, instead of requesting disclosure of training data from the client, we can derive an inner product $\phi(x)^T\phi(w_y)$ between an input $x \in \mathcal{X}$ and a label $w_y \in \mathcal{X}$ for $y \in \{\pm 1\}$, where $\mathcal{X}$ is a set of strings, and $\phi$ represents text embeddings, and we can directly insert desired labels such as $\phi(x)^T[\phi(\text{"positive"}) - \phi(\text{"negative"})]$. Though it seems too straightforward, one can find out that the accuracy for SST-2 is 0.83[1]. The inner product $k(x, w_y) = \phi(x)^T\phi(w_y)$ measuring the text similarity of the sentence pair $k : \mathcal{X}^2 \to \mathbb{R}$ is called a *kernel*.

---

[1] Confirmed through text-embedding-ada-002. The chance rate is 0.50 and the state-of-the-art is 0.95 (fully-trained RoBERTa-large-FT, *i.e.,* supervised learning with GPUs) (Gao et al., 2020).

Here we use a simple string of "positive"/"negative" as an example, but there are countless such task description expressions, each with slightly different vector representations (*e.g.*, "good"/"bad" or "I loved this movie!"/"It was a waste of time."). Therefore, we generalize the above method into a set of $n$ synonyms per label $\{x_1, \ldots, x_{2n}\} = \{w_y^1, \ldots, w_y^n\}_{y \in \{\pm 1\}}$ with a decision function $f_\alpha : \mathcal{X} \to \mathbb{R}$ as follows:

$$f_\alpha(x) = \sum_{i=1}^{2n} \alpha_i k(x, x_i) y_i, \tag{1}$$

where $\alpha_i \geq 0$ represents the *importance* of each description.

The problem is how to find the solution of the system $\alpha$ with $2n$ degrees of freedom without auxiliary oracles such as real labeled samples. Surprisingly, the optimal solution is found by "kernel trick" (Cortes & Vapnik, 1995), solving the dual objective of SVMs (Cortes & Vapnik, 1995) as follows:

$$\max_\alpha J(\alpha) = \sum_{i=1}^{2n} \alpha_i - \frac{1}{2} \sum_{i=1}^{2n} \sum_{j=1}^{2n} \alpha_i \alpha_j k(x_i, x_j) y_i y_j \quad \text{s.t.} \quad \sum_{i=1}^{2n} \alpha_i y_i = 0, \quad 0 \leq \alpha_i \leq C \tag{2}$$

where $C > 0$ is a regularization parameter that controls the trade-off between maximizing the margin and minimizing the training error. This technique also works for any other kernel machine such as ridge regression and Gaussian process and multi-class problems as long as the loss is convex, as stated in the following theorem.

**Theorem 3.1.** (informal; see Appendix.) For a string set $\mathcal{X}$ and its subset $(\mathcal{Y}, \texttt{cls}), \mathcal{Y} \subset \mathcal{X}, \texttt{cls} : \mathcal{Y} \to \{1, \ldots, M\}$, the solution to a convex-optimization problem for a functional $f(\cdot; \mathcal{Y}) : \mathcal{X} \to \mathbb{R}$

$$\min_f \sum_{y \in \mathcal{Y}} \mathcal{L}(f(y; \mathcal{Y}), \texttt{cls}_y) + C^{-1} \|f\|_\infty, \quad C > 0 \tag{3}$$

is represented as $f^* \in \texttt{span}\{k(\cdot, y)\}_{y \in \mathcal{Y}}$ with the existence of a positive-definite kernel $k(\cdot, \cdot) : \mathcal{X}^2 \to \mathbb{R}$.

**Theorem 3.2.** (informal. see Appendix.) If a function $k : \mathcal{X}^2 \to \mathbb{R}$ can be decomposed as $k(x_i, x_j) = \kappa(\phi(x_i), \phi(x_j))$ with the existence of a positive definite function $\kappa : \mathcal{Z}^2 \to \mathbb{R}$ and a function $\phi : \mathcal{X} \to \mathcal{Z}$ in an Euclidean space $\mathcal{Z}$, then $k$ is also a positive-definite kernel and satisfies the conditions of a kernel.

One of the benefits of using a kernel approach is that it can implicitly emulate a higher-dimensional space with low-dimensional kernels. Although not all classification problems on $\phi(\mathcal{X})$ can be linearly separable, from the theorem above, a non-linear kernel beyond an inner product can be defined. For any (possibly non-linear) positive definite function $\kappa(\cdot, \cdot) : \mathbb{R}^{2d} \to \mathbb{R}$ and text embedding $\phi(\cdot) : \mathcal{X} \to \mathbb{R}^d$, we define a *language kernel* as follows:

$$k_\phi(x_1, x_2) := \kappa(\phi(x_1), \phi(x_2)). \tag{4}$$

For example, composition of well-known positive definite kernels such as the RBF and polynomial:

$$k_\phi^{\text{RBF}}(x_1, x_2) = \exp\left(\gamma \|\phi(x_1) - \phi(x_2)\|^2\right), \quad k_\phi^{\text{poly}}(x_1, x_2) = \left(1 + \phi(x_1)^T \phi(x_2)\right)^{d_0} \tag{5}$$

also satisfy the conditions of kernels, and the implicit representation space is of higher dimension. The number of support vectors $n_{\text{SV}} < 2n$ is bounded by the VC-dimension (Vapnik & Chervonenkis, 1971; Blumer et al., 1989), which measures the complexity of a binary classification problem with the maximum number of samples the classifier can shatter.

Is it possible to fully automate the process of sequentially coming up with a text description of the target task, as we often do? This motivates us to consider Support Vector *Generation* (SVG).

## 4 SUPPORT VECTOR GENERATION

The zero-shot decision function in Eq. (1) is represented as follows:

$$f_\alpha(x_{\text{new}}) = \mathbb{E}_{x, y \sim \pi}[k(x_{\text{new}}, x) y], \tag{6}$$

where the $\pi$ is a probability distribution with countable spikes on $\mathcal{X} \times \{\pm 1\}$, each spike having an appropriate ordered index $i$. Here, if we denote the ordered set by $\mathcal{D}$, $\pi$ can be expressed as follows:

$$\pi(x,y) = \begin{cases} \alpha_i / \|\alpha\|_1, & \text{if } (x,y) \in \mathcal{D}, x = x_i, y = y_i \\ 0, & \text{if } (x,y) \notin \mathcal{D} \end{cases} \tag{7}$$

where $\|\alpha\|_1 := \sum_i \alpha_i$ is the $L^1$ norm of $\alpha$. Note that even if $(x,y) \in \mathcal{D}$, if $x$ does not support the decision boundary *i.e.*, $\alpha_i = 0$, then the case is equivalent to $(x,y) \notin \mathcal{D}$. Thus, when we denote the set of support vectors, *i.e.*, the vectors with $\alpha_i > 0$, as $\mathcal{D}_{\text{SV}}$, the $\mathcal{D}$ can be identified with $\mathcal{D}_{\text{SV}}$ (a.c.). As the optimal $\alpha$ is found by minimizing Eq. (2), it is sufficient to find the $\mathcal{D}_{\text{SV}}$ (neither $\mathcal{D}$ nor $\theta$!) to represent the problem. This trick binds the optimal hyperplane to a VC-dimension ($\approx |\mathcal{D}_{\text{SV}}|$), which is finite and smaller than $\dim \theta$, especially for deep neural nets, not to mention that $|\mathcal{D}|$, and then makes the problem far easier than fine-tuning and training data generation.

The idea of SVG is to sample from the prior $p_\theta(x,y)$ and optimize the decision boundary with the kernel machines. However, directly sampling from $p_\theta$ is not easy due to intractability, as in many cases, we only know a scaled and/or conditional $q_\theta(x|y)$ [2]. To address this, we employ Markov chain Monte Carlo (MCMC), particularly Metropolis-Hastings (MH) sampling (Chib & Greenberg, 1995), assuming $\mathcal{D}$ is an ergodic process $x_1 \to x_2 \to \cdots x_t \to \cdots \mid \theta$ whose empirical distribution $p(x,y|\mathcal{D}^{(<t)})$ converges to $\pi(x,y)$ ($t \to \infty$, a.c.).

MH aims to achieve the *detailed balance* on the state transition at the fixed point

$$\pi(x_t, y_t) q_\theta(x_{t+1}|x_t) = \pi(x_{t+1}, y_{t+1}) q_\theta(x_t|x_{t+1}),$$

which is tractable because the normalization of both sides are canceled. Given a sample $x_t$ at each step $t$, the sampling step of MH proposes a new sample from a distribution $q_\theta(x_t \to x_{\text{new}}) = q_\theta(x_{\text{new}}|x_t)$, and decides whether to update $x_{t+1}$ with $x_{\text{new}}$ or $x_t$ based on the following acceptance probability (Chib & Greenberg, 1995) [3]

---

**Algorithm 1** SVG

**Require:** $x_1, x_2, \phi, \theta, n, C$
1: $\mathcal{D} \leftarrow \{(x_1, +1), (x_2, -1)\}; \alpha \leftarrow \mathbf{1}C$
2: **for** $t = 2$ to $2n$ **do**
3: $\quad x_{\text{new}} \sim q_\theta(\cdot|x_t)$
4: $\quad y_{\text{new}} \leftarrow \text{sgn} f_\alpha(x_{\text{new}})$ on $\mathcal{D}, \alpha, k_\phi$
5: $\quad \mathcal{D}' \leftarrow \mathcal{D} \cup \{(x_{\text{new}}, y_{\text{new}})\}$.
6: $\quad \alpha' \leftarrow \arg\max J(\cdot)$ on $\mathcal{D}', C, k_\phi$
7: $\quad u \sim \text{Uniform}(0,1)$
8: $\quad$ **if** $u < \tilde{A}_{t+1}(x_t, x_{\text{new}})$ (Eq. 9) **then**
9: $\quad\quad \mathcal{D} \leftarrow \mathcal{D}'; \quad \alpha \leftarrow \alpha'$
10: $\quad\quad (x_{t+1}, y_{t+1}) \leftarrow (x_{\text{new}}, y_{\text{new}})$
11: $\quad$ **else**
12: $\quad\quad (x_{t+1}, y_{t+1}) \leftarrow (x_t, y_t)$
13: $\quad$ **end if**
14: **end for**
**Ensure:** $\mathcal{D}_{\text{SV}} \leftarrow \{(x_i, y_i) \in \mathcal{D} | \alpha_i > 0\}$.

---

$$A(x_t, x_{\text{new}}) = \min\left[1, \frac{\pi(x_{\text{new}}, y_{\text{new}})\, q_\theta(x_t|x_{\text{new}})}{\pi(x_t, y_t)\, q_\theta(x_{\text{new}}|x_t)}\right]. \tag{8}$$

We estimate the backward using $q_\theta(x_t|x_{\text{new}}) = q_\theta([x_{\text{new}}; x_t])/q_\theta(x_{\text{new}})$, where $[x_{\text{new}}; x_t]$ is the concatenation. As $\pi$ is unknown, we approximate it with a scaled $\tilde{\pi}_{t+1}(x_{\text{new}}, y_{\text{new}}) = \max(0, \alpha'_{t+1} y_{\text{new}} f_{\alpha'}(x_{\text{new}}))$, where $\alpha'$ is the dual coefficient learnt assuming $(x_{t+1}, y_{t+1}) = (x_{\text{new}}, \text{sgn} f_\alpha(x_{\text{new}}))$. As by definition, $y_{\text{new}}$ takes either binary value, we can write $y_{\text{new}} = \text{sgn} f_{\alpha'}(x_{\text{new}})$ a.c. without loss of generality. With $\tilde{\pi}_{t+1}$, the above equation is approximated as,

$$\tilde{A}_{t+1}(x_t, x_{\text{new}}) = \min\left[1, \frac{\alpha'_{t+1}\, q_\theta([x_{\text{new}}; x_t])\, q_\theta(x_t)}{\alpha'_t\, q_\theta([x_t; x_{\text{new}}])\, q_\theta(x_{\text{new}})}\right]. \tag{9}$$

The proposed SVG algorithm is outlined in Algorithm 1. This method generates additional support vectors that help shape decision boundaries achieved by sampling from the prior distribution and

---

[2] Though completion models provide the transition $p_\theta(x_{\text{completion}}|x_{\text{prompt}}) = \prod_{i=1}^m p_\theta(w_i|x_{\text{prompt}}, w_{<i})$, the stable probability $p_\theta(x_{\text{completion}})$ is intractable: $\int_{\mathcal{X}} p_\theta(x_{\text{completion}}|x_{\text{prompt}}) dp_\theta(x_{\text{prompt}})$. This problem, concerning the "initial token" $p_\theta(x)$, is related to symbol grounding and multi-modality.

[3] Intuitively, the weight $q_\theta(x_{t+1} \to x_t)/q_\theta(x_t \to x_{t+1})$ penalizes when the backward path $x_{t+1} \to x_t$ is too low. For example, in text generation, "lions are $\to$ mammals" is correct, but "mammals are $\to$ lions" are not always correct or depends on contexts, so $q_\theta(\text{"mammals"}|\text{"lions"})/q_\theta(\text{"lions"}|\text{"mammals"})$ should be low. We can confirm that $A = 0$ if $f_t(x_{t+1})$ is misclassified or $A = 1$ if the process satisfies the detailed balance for backward/forward sampling $q_\theta(x_{t+1}|x_t) = q_\theta(x_t|x_{t+1})$.

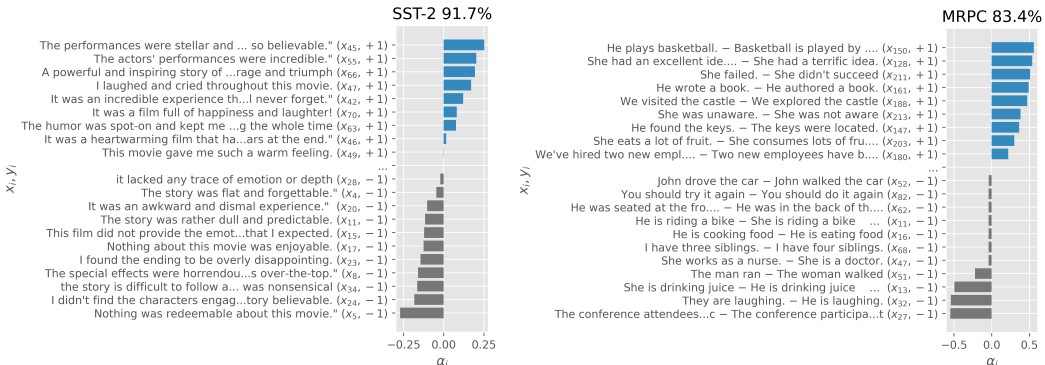

Figure 1: The spikes — high-performed support vectors generated from the SVG algorithm. The vectors are visualized from a chain out of five parallel MCMC search.

subsequently updating the parameters of the kernel machine. The advantage of this method is that it allows artificial expansion of the training data, which is particularly useful in scenarios where available data is scarce. By encapsulating the data distribution more effectively, the SVG ensures improved performance of zero-shot learning models, even under computational constraints. Table 1 illustrates the complexity analysis. Even though the QP solver for Eq. (2) takes $O(n^3)$ times, we assume the practical algorithm for SVMs such as Sequential Minimal Optimization (SMO) (Platt, 1998), which takes $O(n^2)$ times. In *few*-shot learning, SVG is faster than the Transformers because $n \ll m^2$. The complexity of LLMs are borrowed from the original paper of the Transformer (Vaswani et al., 2017).

## 5 NUMERICAL EXPERIMENT

To evaluate the effectiveness of our proposed approach, we conducted experiments on the General Language Understanding Evaluation (GLUE) benchmark (Wang et al., 2019a). The GLUE benchmark comprises a set of sentence or sentence-pair language understanding tasks, providing a comprehensive evaluation of the performance of language models in various natural language understanding scenarios. We have compared the performance of our proposed method, SVG to a baseline methodology: 'Prompting' (Brown et al., 2020). In this comparison, we employed a conventional prompting technique, a zero-shot learning approach that adopts a set of manually-constructed prompts as an exemplar for the labels. For context, we used a non-fine-tuned LLM (Gao et al., 2020) comprising our baseline.

Table 1: Complexity analysis of SVG and LLMs. $n$: the number of samples, $m$: the maximum length of texts in token, $d$: the dimension of the embeddings.

|  | Train | Predict |
|---|---|---|
| SVG | $O(n^2 \cdot d)$ | $O(n \cdot d)$ |
| LLMs (fine-tuning) | $O(n \cdot m^2 \cdot d)$ | $O(m^2 \cdot d)$ |

The experimental configuration used two CPU-only virtual machines on a public cloud as the computational environment and an OpenAI pay-as-you-go account as the trained language model. Three executions per task were carried out in a multi-process manner, with one CPU (not GPU) of 3 GHz and 1 GB memory are assigned to each process. The training was completed in three minutes, which is far faster and more economical than networks with GPUs.

### 5.1 RESULTS

The experimental results are shown in Table 2. We report the accuracy and F1 score for each task, comparing the performance of SVG with the baseline methods. The results show that SVG outperforms the baseline methods in terms of both accuracy and F1 score, demonstrating the effectiveness of SVG in improving the performance of zero-shot learning tasks, even in resource-constrained environments. Fig. 1 and 2 shows the generated samples and the decision boundary.

Table 2: Results from zero-shot learning of GLUE benchmark with CPUs and without any GPUs or GPU memories. The values of † are from Gao et al. (2020). The experiment was repeated three times, and the average and standard deviation are listed. **Bold** indicates the best score. The rightmost column shows the average elapsed time for a single experiment.

| | Single Sentence | | Sentence-pair | | | | | |
|---|---|---|---|---|---|---|---|---|
| | **SST-2** (Acc.) | **CoLA** (Matt.) | **QQP** (F1) | **MRPC** (F1) | **RTE** (Acc.) | **QNLI** (Acc.) | **MNLI** (3-Acc.) | (sec) |
| Chance rate | *50.0* | *0.0* | *50.0* | *50.0* | *50.0* | *50.0* | *33.3* | - |
| SVG (ours) | **91.7**$_{0.9}$ | **9.1**$_{3.2}$ | **72.9**$_{1.5}$ | **63.7**$_{12.1}$ | **57.9**$_{2.8}$ | **64.0**$_{5.2}$ | 51.8$_{1.2}$ | 48.1 |
|   without MCMC | 88.9$_{1.5}$ | 4.3$_{1.5}$ | 65.7$_{2.1}$ | 58.1$_{8.0}$ | 55.1$_{1.2}$ | 55.9$_{4.8}$ | 44.9$_{1.2}$ | **1.9** |
| Prompting† | 83.6$_{0.0}$ | 2.0$_{0.0}$ | 49.7$_{0.0}$ | 61.9$_{0.0}$ | 51.3$_{0.0}$ | 50.8$_{0.0}$ | 51.7$_{0.0}$ | - |

The superior performance of SVG can be attributed to the combination of the generative capabilities of LLMs and the computational efficiency of kernel machines. By generating support vectors from the LLMs, SVG is able to augment the training data for zero-shot learning tasks without the need for fine-tuning or additional computational resources. This allows SVG to achieve comparable or even superior performance to the baseline methods, while using only CPU resources.

The experimental results overall corroborate the effectiveness of SVG in zero-shot learning, specifically in resource-constrained contexts. Merging LLMs with kernel machines introduces new opportunities for elaborated language processing assignments. Such a combination facilitates the creation of precise and efficient natural language understanding systems.

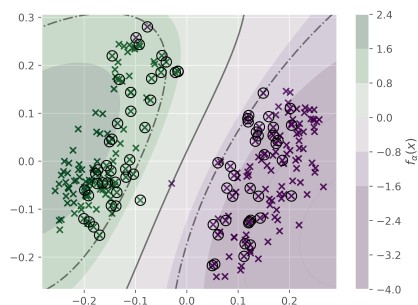

Figure 2: Generated samples and the decision boundary, with the points in circles representing support vectors and contours with $f_\alpha(x) = +1, 0, -1$ from the left. SST-2 (Accuracy: 0.917), text-curie-001.

# 6 DISCUSSION

## 6.1 APPLICATIONS

While we have mainly discussed the case of single sentences with two classes, in tasks of natural language understanding, we often have to deal with more than two classes and more than one sentence. GLUE benchmark has sentence-pair classification tasks such as paraphrase identification (QQP, MRPC) and inference (RTE, QNLI), and multi-class tasks (MNLI) where the model has to predict one of three classes (entailment, neutral, contradiction) for a sentence pair. Each of the tasks also has a training data which has not been covered yet in this paper, though application to few-shot learning is also possible. In this section, we discuss how to extend the proposed method to such tasks.

### 6.1.1 SENTENCE-PAIRS

For sentence-pair classification, we can adopt the same methodology as in the case of single-sentence classification, albeit with a few alterations. Initially, our task is to represent the features of the sentence-pair. We can accomplish this by utilizing a language kernel $k_\phi : \mathcal{X}^4 \to \mathbb{R}$, which accepts two sentences as input and is defined as follows[4]

$$k_\phi([x_1; x_2], [x_3; x_4]) = \kappa(\phi(x_2) - \phi(x_1), \phi(x_4) - \phi(x_3)), \tag{10}$$

where $\phi : \mathcal{X} \to \mathbb{R}^d$ denotes a text embedding and $\kappa : \mathbb{R}^{2d} \to \mathbb{R}$ represents a positive definite kernel.

---

[4] We have assumed a recurring topology $\mathcal{X}^2 \subset \mathcal{X} = \mathcal{V}^m$, *i.e.*, pairings of sentences constitute a language just as individual sentences do, hence $k_\phi : (\mathcal{X}^2)^2 \to \mathbb{R}$ also constitutes a language kernel. Following this assumption, the language kernel can be alternatively written as $k_\phi : \mathcal{X} \to \mathbb{R}$. This topic is slated for discussion in future work.

Second, we need to compute the acceptance probability $\tilde{A}_{i+1}(x_i, x_{\text{new}})$ for a sentence-pair. To do this, we can use the same approach as for single-sentence classification, but with a few modifications. First, we need to compute the probability $q_\theta(x_{\text{new}}|x_i)$ for the sentence-pair. To do this, we can use a language model $q_\theta : \mathcal{X}^2 \to \mathbb{R}$ that takes two sentences as input defined as $q_\theta([x_1; x_2]) = \frac{1}{Z} \exp\left(\sum_{i=1}^2 \log q_\theta(x_i)\right)$, where $Z$ is a normalization constant and $q_\theta(x_i)$ is a language model for a single sentence. Finally, we can compute the acceptance probability $\tilde{A}_{i+1}([x_i^1; x_i^2], [x_{\text{new}}^1; x_{\text{new}}^2])$ for a sentence-pair using $q_\theta$.

### 6.1.2 MULTI-CLASS

In multi-class classification tasks, we have to deal with more than two classes. Typically, we can use the one-vs-rest (OVR) or one-vs-one (OVO) approach. The former constructs a binary classifier for each class, and the class with the highest score is chosen as the predicted class, whereas the latter constructs a binary classifier for each pair of classes, and the class with the highest number of wins is chosen as the predicted class.

For the OVR approach, we can use the same algorithm as for the binary classification case, with the only difference being that the labels $y_i$ are now one-hot vectors instead of the binaries $\{\pm 1\}$. For OVO, we can use the same algorithm, but with the labels $y_i$ being a vector of length $M(M-1)/2$, where $M$ is the number of classes. To calculate the acceptance ratio in the MCMC step, we need to compute the backward probability $q_\phi(x_i|x_{\text{new}})$. For OVR, we can simply use the same formula as for the binary classification case. On the other hand, for OVO we need to compute the backward probability for each pair of classes.

### 6.1.3 FEW-SHOT LEARNING

Although we can simply initialize $\mathcal{D}$ with the given labeled samples from the training set and optimize $\alpha$ with Eq. (2), SVG also retains the ability to generate additional support vectors that resemble these labeled samples. This is primarily because these generated support vectors are expected to be located around the decision boundary, indicating that they should closely mirror the given labeled samples. Where this approach deviates from data augmentation is in its ability to generate samples which are more semantically similar to the given labeled samples than the samples generated by data augmentation. This can be attributed to the fact that SVG is based on a kernel machine, a method that effectively captures the similarity between samples by transforming the input data into a high-dimensional space, enabling a more nuanced similarity measure.

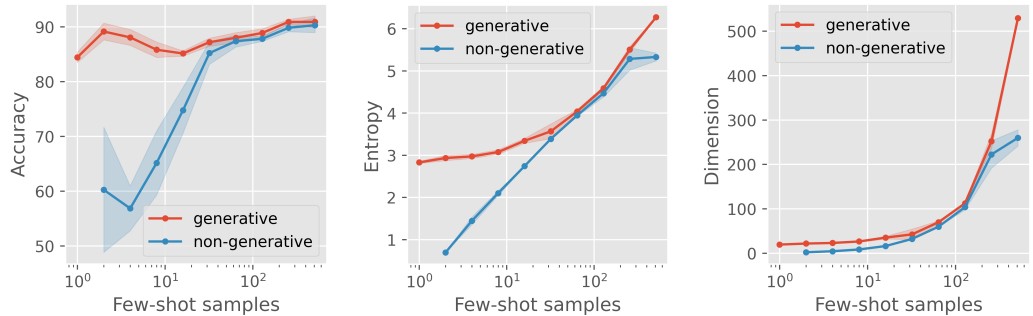

Figure 3: The result of few-shot learning of SVG through SST-2 in comparison to the conventional kernel machines, such as non-generative SVMs. The accuracy shows that even for the scarce dataset, SVG can complement the lack of data points. Entropy $\mathbb{H}[\tilde{\pi}_n] := \log \|\alpha\|_1 - \sum_{i=1}^n (\alpha_i/\|\alpha\|_1) \log \alpha_i$ shows that SVG successfully obtains the complexity for the scarce dataset. The right figure shows the number of support vectors acquired by each method.

Fig. 3 shows the results of few-shot learning. In machine learning theory, including few-shot learning, that accuracy increases with the number of data. Interestingly, in the case of SVG, accuracy reaches a maximum when the number of data is two (*i.e.,* one positive example and one negative example each), then drops and steadily increases again when the number of train data is sufficient.

One possible explanation for this result is that while the generalisation performance is improved by incorporating different distributions as long as the number of 'external' data to be added is small, the quality of the external data is inferior to the quality of the train data generated by the SVG itself, which in turn hurts the accuracy of the model. This phenomenon is referred to as 'few-shot double descent' in SVG, though the reasons for this are not analysed further in this paper.

## 6.2 REPRODUCIBILITY

### 6.2.1 MODEL SELECTION

Performance of kernel machines highly depends on the hyper-parameter $C$, the upper bound of each entry of $\alpha$, and is on a trade-off between over- and underfitting (Duan et al., 2003). If $C$ is too large, the model yields overfitting, and *vice versa*. One of the common search algorithms is the combination of grid search and cross-validation (Syarif et al., 2016): we divide $\mathcal{D}$ into $K$ discrete batches randomly and test the each with $K - 1$ others, and choose the best candidate maximizing the metrics, as shown in Fig. 4. We optimized $C = C_0/n$ for the number of samples $n$ and select the best candidate $C_0$ from a discrete log space $\{10^{-2}, \ldots, 10^{10}\}$ at every $t_0 = 10$ step.

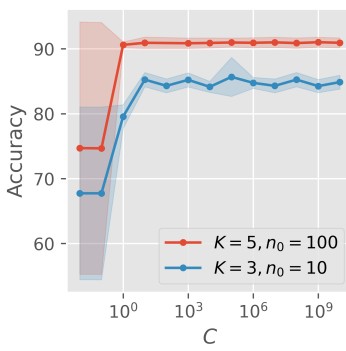

Figure 4: Results of performance by fixed $C$ on SST-2.

A proposal $q_\theta$ as close as the target $\pi$ improves the acceptance ratio of MH. Other than completion API as we have employed in this paper, there are several heuristics of data augmentation such as back translation and text attack (Brislin, 1970; Morris et al., 2020).

### 6.2.2 VARIANCE REDUCTION

As we approximate the target probability $\pi$ with $\tilde{\pi}_t$, there is a risk of exponential amplification of the model's approximation error as the generative progresses, especially if the dynamics $\mathcal{T} : \tilde{\pi}_t \rightarrow \mathcal{T}\tilde{\pi}_t$ is non-contractive in the measurable space on $\mathcal{X}$. To mitigate this drawback, we employ a "cross-validating" approach for posterior estimation in MCMC (Held et al., 2010). We run $K$ independent chains in parallel, and use the posterior approximation $\tilde{\pi}_t^{i-1}$ from the previous chain as an approximation to $\pi$ for the current chain, instead of using $\tilde{\pi}_t^i$ directly. Additionally, we have introduced the following heuristics into the implementation:

- A burn-in period $t_0$, assuming that the variance of $\tilde{\pi}_t$ stabilizes after $t_0$ iterations. We update the reference model $\tilde{\pi}_t$ every $t_0$ steps, instead of updating it at every step, to reduce the sensitivity of the initial samples and improve the overall stability of the approximation.

Table 3: The model-agnostic multi-class task descriptions $x_1, x_2[, \ldots, x_M]$ which yield high performance in SVG. The placeholder of quote, labels and sample are obtained from the GLUE original paper (Wang et al., 2019a), which can also be scraped at https://tensorflow.org/datasets/catalog/glue. Instead of text-curie-001 for MCMC, the initial samples will be inferred once via larger completion models such as text-davinci-003 .

**(a) Template**

```
<quote>
  1: <label₁>, 2: <label₂>, ..., M: <label_M>
```
The possible ten examples of the `<sample>` of "$i$: `<label_i>`" are:

**(b) SST-2** (single sentence, 2 classes)

"The Stanford Sentiment Treebank consists of sentences from movie reviews and human annotations of their sentiment. The task is to predict the sentiment of a given sentence. We use the two-way (positive/negative) class split, and use only sentence-level labels."

1: positive, 2: negative

The possible ten examples of the sentence of "2: negative" are:

- We incorporate a probabilistic SVM (Wu et al., 2003) and multiply the probability $p(y_t|x_{new})$ to $\pi_t$. This helps to refine the estimation of the posterior by incorporating the predictive power of the SVM.

- If the newly generated samples $x_{\text{new}}$ duplicates any of the previous ones, or if is one of the predefined stop words *e.g.*, ".", `<EOS>`, or `<LF>`, we set $\alpha'_{t+1} = 0$ and generate a new token.

To address the issue of sensitivity to initial samples, we employ transfer learning for initial sampling. We directly draw $2n_0$ seeds from the task description in the original paper of the target dataset (Wang et al., 2019a). We carefully select task descriptions that are model-agnostic and unbiased, enabling us to train our model in a zero-shot manner. While we use a mid-size completion model text-curie-001 during the process, we utilize text-davinci-003 for the initial sampling. We have observed that repeating this initial sampling process also yields high performance, but it increases the overall computational cost compared to MCMC. Therefore, using this heavy sampling approach once at $t = 0$ is the most practical option. In Table 3, we provide examples of the model-agnostic task descriptions used for transfer learning. These examples demonstrate the effectiveness of our approach in achieving high performance across different tasks. An exhaustive list of models and hyperparameters used in the experiments of this paper is presented in Table 4.

### 6.3 LIMITATION

While our proposed approach has yielded promising results, it presents certain limitations. First, SVG's effectiveness hinges on the quality of the representative samples that the LLMs generate. The interpretability of these generative models often lacks clarity, and they may inadvertently perpetuate undesirable biases inherent in the training data (Zhao et al., 2018).

Second, in spite of superior memory efficiency compared to contemporary LLMs, the computational efficiency of kernel methods still tends to deteriorate significantly when dealing with high-dimensional embeddings or many-shot problems. As for computational speed $O(n^2)$, kernel methods frequently lag behind LLMs $O(nm^2)$ of $n \gg m^2$.

We anticipate addressing this limitation in our future work. One proposed direction of research involves the development of more informed and reliable strategies for generating representative

Table 4: Models and hyperparameters used for GLUE benchmark. These are shared by all the tasks after tuned with SST-2.

|  | Description | Value |
|---|---|---|
| $\phi$ | A text embedding | text-embedding-ada-002[†] |
| $\theta$ | Language model | text-davinci-003[†] (seeds) text-curie-001[†] (MCMC) |
| $f_\alpha$ | Kernel machine | libsvm[‡] |
| $n$ | Examples per class | 100+1000 (seeds/MCMC) |
| $d$ | Dimension of the embeds | 1536[†] |
| $m$ | Length in tokens | 2048[†] |
| $C_0$ | Upper bound of the duals $\alpha_i$ | $10^{-2}, 10^{-1}, \ldots, 10^{10}$ |
| $\kappa$ | A sub-kernel | RBF, Linear |
| $\gamma$ | Scaler of the RBF kernels | auto[‡] |
| $K$ | Chains in parallel. | 5 |
| $t_0$ | Burn-in period | 10 |
|  | Multi-class settings | OVR, OVO |
|  | Cross-validating bins | 5 |
|  | Temperature | $.5 \pm .05$ |
|  | Prompt | $x$ as _? |
|  | Stop words | `<EOS>`|`<LF>`|. |

[†]GPT-3 (Brown et al., 2020) [‡]Chang & Lin (2011)

samples. This could mean leveraging insights from active learning or using novel architectures that encourage diversity in the generated examples. We also aim to explore efficient kernel methods and applicable approximations, such as random Fourier features (Rahimi & Recht, 2008), to tackle large data scenarios. Pursuing these avenues, we believe, will bring us closer to creating efficient and effective zero-shot learning models with a wider range of applicability.

**Conclusion** In this paper, we proposed a novel approach for zero-shot learning, Support Vector Generation (SVG), which combines the generative capabilities of LLMs with the computational thriftiness of kernel machines. We demonstrated the mathematical equivalence of zero-shot learning and kernel machines, and showed that our approach can be used to augment the data for zero-shot tasks without the need for train data nor high-performance computing resources. Our experiments on GLUE showed that SVG can achieve competitive performance compared to existing methods. Our contribution introduces a fresh perspective on zero-shot learning, presenting a potential answer to the challenge posed by resource-limited environments. In our view, it is conceivable that our approach might be generalized to incorporate tasks such as one-class problems and regression. We anticipate our work will stimulate subsequent investigations in this domain.

## ETHICAL CONSIDERATIONS[5]

As we embark on this journey of leveraging the power of Large Language Models (LLMs) and kernel machines, it is essential to consider the ethical implications that may arise.

Firstly, the use of LLMs, while impressive in their performance, have been known to potentially propagate and reinforce existing biases in the data they are trained on (Bolukbasi et al., 2016). This can result in biased outputs, which can be harmful and unfair, particularly when these models are used in sensitive applications such as hiring, loan approvals, or law enforcement. As part of our commitment to responsible AI, we must ensure that our proposed methods do not exacerbate these biases and that we take steps to mitigate them where possible.

Secondly, the accessibility of these models in resource-constrained environments is a double-edged sword. While it allows for the democratization of AI, enabling more people to benefit from these technologies, it also raises concerns about misuse. For instance, these models could be used to generate misleading information or propaganda, or to automate cyber attacks. Therefore, it is crucial to establish robust guidelines and safeguards to prevent misuse.

Lastly, the generation of training data for zero-shot learning tasks must be handled carefully to ensure that any data used for this purpose is anonymized and that individuals' privacy is respected. Generative classifiers can have the paradox that the generative model itself is included among the entities that generate harmful speech, such as hate speech and slander, when the MCMC posterior distribution approximation is used to automatically detect sentences of anti-social speech. One aspect of SVG that makes it useful from a responsible AI perspective is that it minimizes the generation of extreme sentences, as the generated samples are only distributed around the classification boundaries. SVG's sparsity is guaranteed by the hinge loss minimization and margin maximization principles of kernel machines, which automatically eliminate samples that are merely harmful and do not contribute to classification. Although SVG by itself is not a safety-conscious mechanism, it is a direction for future research. As a future research direction, we are also considering the construction of classifiers that do not generate anomalies themselves, for example by utilizing one-class SVMs.

In conclusion, our proposed method holds great promise for advancing the field of natural language understanding. However, it is essential that we navigate this path with a keen awareness of the potential ethical implications. As we continue to develop and refine our approach, we must ensure that we uphold the principles of fairness, accountability, transparency, and respect for user privacy.

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

## A  PROOF OF THEOREM 3.1–3.2

**Theorem 3.1.** For a string set $\mathcal{X}$ and its subset $(\mathcal{Y}, \texttt{cls}), \mathcal{Y} \subset \mathcal{X}, \texttt{cls} : \mathcal{Y} \to \{1, \ldots, M\}$, the solution to a convex-optimization problem for a functional $f(\cdot; \mathcal{Y}) : \mathcal{X} \to \mathbb{R}$

$$\min_f \sum_{y \in \mathcal{Y}} \mathcal{L}(f(y; \mathcal{Y}), \texttt{cls}_y) + C^{-1}\|f\|_\infty, \quad C > 0 \tag{11}$$

is represented as $f^* \in \texttt{span}\{k(\cdot, y)\}_{y \in \mathcal{Y}}$ with the existence of a positive-definite kernel $k(\cdot, \cdot) : \mathcal{X}^2 \to \mathbb{R}$.

*Proof.* We firstly note that the optimization problem is convex and the solution $f^*$ is unique. Let $f^*$ be the solution of the optimization problem. Then, $f^*$ can be written as a linear combination of the kernels $k(\cdot, y)$ with some coefficients $\alpha_y$ for $y \in \mathcal{Y}$ as follows,

$$f^*(x) = \sum_{y \in \mathcal{Y}} \alpha_y^* k(x, y). \tag{12}$$

We then show that the coefficients $\alpha_y$ are positive. Let $f_\alpha$ be a function defined as follows,

$$f_\alpha(x) = \sum_{y \in \mathcal{Y}} \alpha_y k(x, y). \tag{13}$$

Then, we have

$$\|f_\alpha\|_\infty \leq \sum_{y \in \mathcal{Y}} |\alpha_y| \|k(\cdot, y)\|_\infty \leq C. \tag{14}$$

Since $f^*$ is the solution of the optimization problem, we have

$$\sum_{y \in \mathcal{Y}} \mathcal{L}(f^*(y; \mathcal{Y}), \mathtt{cls}_y) \leq \sum_{y \in \mathcal{Y}} \mathcal{L}(f_\alpha(y; \mathcal{Y}), \mathtt{cls}_y). \tag{15}$$

By the convexity of the loss function $\mathcal{L}$, we have

$$\mathcal{L}(f^*(y; \mathcal{Y}), \mathtt{cls}_y) \leq \mathcal{L}(f_\alpha(y; \mathcal{Y}), \mathtt{cls}_y) \leq \sum_{y \in \mathcal{Y}} |\alpha_y| \mathcal{L}(k(x, y), \mathtt{cls}_y). \tag{16}$$

Since the loss function $\mathcal{L}$ is bounded from below, we have

$$\sum_{y \in \mathcal{Y}} |\alpha_y| \geq \frac{1}{\mathcal{L}_{\min}} \sum_{y \in \mathcal{Y}} \mathcal{L}(f^*(y; \mathcal{Y}), \mathtt{cls}_y). \tag{17}$$

Therefore, the coefficients $\alpha_y$ are positive.

Finally, we show that the kernel $k(\cdot, \cdot)$ is positive-definite. Let $x_1, \ldots, x_n \in \mathcal{X}$ be arbitrary points. Then, we have

$$\sum_{i,j=1}^n \alpha_i \alpha_j k(x_i, x_j) = \sum_{i,j=1}^n \alpha_i \alpha_j \sum_{y \in \mathcal{Y}} k(x_i, y) k(x_j, y) \geq 0. \tag{18}$$

Therefore, the kernel $k(\cdot, \cdot)$ is positive-definite. This completes the proof.

**Theorem 3.2.** If a function $k : \mathcal{X}^2 \to \mathbb{R}$ can be decomposed as $k(x_i, x_j) = \kappa(\phi(x_i), \phi(x_j))$ with the existence of a positive definite function $\kappa : \mathcal{Z}^2 \to \mathbb{R}$ and a function $\phi : \mathcal{X} \to \mathcal{Z}$ in an Euclidean space $\mathcal{Z}$, then $k$ is also a positive-definite kernel and satisfies the conditions of a kernel.

*Proof.* Let $x_1, \cdots, x_n \in \mathcal{X}$. Then,

$$\sum_{i=1}^n \sum_{j=1}^n c_i c_j k(x_i, x_j) = \sum_{i=1}^n \sum_{j=1}^n c_i c_j \kappa(\phi(x_i), \phi(x_j)) \tag{19}$$

$$= \sum_{i=1}^n \sum_{j=1}^n c_i c_j \kappa(z_i, z_j) \quad \geq 0 \tag{20}$$

where $z_i = \phi(x_i)$. Therefore, $k$ is a positive-definite kernel.

