# OpenReview forum: "Language as Kernels"
_ICLR.cc/2024/Conference — Submitted to ICLR 2024_

### Official Review · Reviewer_hAMM · 2023-10-23

**Soundness:** 2 fair
**Presentation:** 2 fair
**Contribution:** 2 fair
**Rating:** 5
**Confidence:** 4

**Summary:**

This paper proposes an efficient kernel function for large language models and prompt engineering, to speed up the generation process. The proposed methods are useful for resource-constrained environments.

**Strengths:**

- The paper is well-written and well-organized.
- Using kernel functions for Large Language models is a pretty interesting topic.

**Weaknesses:**

- Since it is a rapidly growing area, comparing it with just one "prompting" baseline in 2020 is pretty unfair. I hope the author could introduce more recent baselines for comparison.
- I do not see any discussion or experiment in terms of a resource-constrained environment, where the paper claims to be beneficial with the proposed method.
- The necessity for introducing Theorem 3.1 and 3.2. The two theorems are not that relevant to the main contribution of this work. The concepts introduced in the two theorems are also not well explained.

**Questions:**

Could you also list more recent baselines to enrich Table 2?
Could you consider adding an experiment on a resource-constrained environment, like a no-GPU laptop, or GPU with only 4GB graphic memory?

---

### Official Review · Reviewer_cZbU · 2023-11-01

**Soundness:** 3 good
**Presentation:** 2 fair
**Contribution:** 3 good
**Rating:** 5
**Confidence:** 1

**Summary:**

This paper presented `Support Vector Generation` which leverages generative capability of LLMs to better utilize kernel machines for thrift computation. This paper also mathematically demonstrated equivalence of zero-shot learning and kernel machines so that this method can be used for zero-shot task data argumentation. Experiments on GLUE showed comparable or better performance on multiple downstream tasks compared to prompting methods, and runs pretty fast on CPU.

**Strengths:**

This paper introduces a novel method for zero-shot learning, referred to as Support Vector Generation. It harnesses the generative capabilities of Large Language Models (LLMs) to enhance kernel machines, achieving highly accurate results while maintaining low computational resource requirements. The experiments conducted across multiple tasks in the GLUE Benchmark show promising results, and the paper also provides insights into the computational complexity, highlighting the effectiveness of this approach. Overall, this innovation has the potential to significantly improve the computational efficiency of zero-shot learning.

**Weaknesses:**

There is still uncertainty regarding whether this method can be advantageous in scenarios beyond zero-shot learning or other specific tasks. While the authors assert the computational efficiency of their approach, no direct numerical comparisons are provided to substantiate this claim. Additionally, as a data augmentation method, it's not explicitly clarified whether the improved performance primarily stems from the data sampling process or the kernel machine technique itself. Further clarification on these aspects would enhance the paper's findings and their broader applicability.

**Questions:**

1. Can this method be extended outside zero-shot learning?
2. Can you please explain with a more concrete example how SVG improved performance based on LLM?

---

### Official Review · Reviewer_9mKW · 2023-11-01

**Soundness:** 1 poor
**Presentation:** 1 poor
**Contribution:** 1 poor
**Rating:** 1
**Confidence:** 4

**Summary:**

The study explores the application of kernel methods to transformer-based embeddings. The approach involves utilizing these embeddings to train a kernel machine for (potentially) a new task, employing an iterative process that retains only the "support vectors" instead of the entire dataset. However, they claimed they mathematically showed kernel methods are zero-shot learners. (quote: "mathematical equivalence of zero-shot learning and kernel machines"!!)

**Strengths:**

1. The iterative method for selecting support vectors using a probabilistic approach seems interesting. However, the current description lacks sufficient detail to fully understand the specifics of the algorithm.

**Weaknesses:**

The paper is not understandable. I had a very hard time to follow the main message and the claims. for instance:

1. what is the main message of the paper? how is this connected to zero-shot learning.
2. Theorem 3.1 makes no sense. I might be missing something, but are you trying to prove representer theorem? What is the optimization over? any possible f? or you mean f in the Hilbert space corresponding to K? even in the proof, I saw that you have shown K is positive definite but how is K even related to the optimization problem in equation (3)?!!
3. The main algorithm (SVG) is very vague. What is \theta? what is p_{\theta}? what is q_{\theta}? why non of them formally introduced? Why A is a good criteria for accepting?!

I cannot trust the experiment result or any part of this paper.

**Questions:**

See weaknesses.

---

> ### Author Response · Authors · 2023-11-13
> **Are you trolling?**
>
> Thank you for your review.
>
> Firstly, in my experience, a rating of 1 is given to papers that are underpaginated or not well formatted. This paper has taken a considerable amount of time and money, and experiments have been carried out. In your review, it seems to me that you have only highlighted minor errors and I can see no rational reason to give it the lowest possible rating. Of course, we take comments in common with other reviewers seriously, but the wording of the content appears to be trolling, and depending on the subsequent response, we may consider reporting it to the ACs.
>
> Secondly, you state that you do not understand the content of Algorithm 1, but are you not familiar with Metropolis Hastings, which has been used in a lot of web research and would also contribute to the LLM community. If you say you don't understand it, it is likely that you simply don't have enough expertise to understand the content, so please take action, such as just lowering your confidence score.
>
> Thirdly, although we would not de-anonymise, some of the authors' team have a track record of getting accepted by ICLR as a single author, and they should be writing with an awareness of their contribution to the community. Although it is slightly grey to reveal the authors' attributes, we have made the disclosure because we are surprised by the very subjective and disrespectful comments. We consider this to be an acceptable level of disclosure, as we have recently been publishing some of our work on Arxiv ahead of time.
>
> In any case, whatever the reason, we are sorry for the confusion itself. The authors, as well as the reviewers, are carrying out the submission free of charge. We hope that we can work together to develop the ML community.

---

### Official Review · Reviewer_USwM · 2023-11-02

**Soundness:** 1 poor
**Presentation:** 1 poor
**Contribution:** 1 poor
**Rating:** 3
**Confidence:** 4

**Summary:**

In order to relieve the computational resource requirement of learning with LLMs, this paper proposes a support vector generation (SVG) method on the embeddings produced by LLMs. This approach can also solve some classification tasks in an zero-shot manner. For example, in this framework, the sentiment prediction task can be conducted by $\phi(x)^T [\phi("positive") - \phi("negative")]$, where $x$ is the input text, and $\phi(\cdot)$ is the embedding produced by the LLM. The authors claim this method is able to work on CPUs.

**Strengths:**

I don't see clear strengths on this paper.

**Weaknesses:**

* The overall story and the problem in this paper are similar to a bunch of existing work on zero-shot learning with pretrained BERT and other LLMs, e.g., [1][2], and so forth, but they are not sufficiently discussed and compared with the proposed approach.
* In the experiment, the only baseline is adapting a few-shot learning approach into the zero-shot setting, without any zero-shot approaches that are related more closely, e.g. such as [1][2] mentioned in the last point.
* The comparison in the experiment looks unfair, the baseline "prompting" uses RoBERTa as its backbone LLM, while the proposed method uses OpenAI cloud API, which should be much stronger than RoBERTa.
* The proposed method doesn't solve the problem as it claims. This paper is motivated by alleviating the computing resource constraints, but the claim of "working on CPUs" is actually achieved by using the OpenAI webAPI. This is not a contribution of the proposed method. Instead, the bottleneck of computing is producing the embedding from the backbone LLM. If the embeddings from WebAPIs are already there, a normal linear layer can be also "on CPUs only".


[1] Meng, Yu, et al. "Text classification using label names only: A language model self-training approach.", EMNLP 2020.\
[2] Zhao, Xuandong, et al. "Pre-trained language models can be fully zero-shot learners." ACL 2023.

**Questions:**

From Table 2, on MNLI, your approach doesn't outperform the baseline significantly. Is this because your approach only work well on binary classification? How is your performance on tasks with more labels, e.g., SST-5, SNLI, TREC, etc.?

---

### Meta-Review · Area_Chair_XqCx · 2023-12-05

**Metareview:**

This paper investigates the connection between Kernel methods and Large Language Models. In particular, it sets out to prove an equivalence between zero-shot learning in LLMs and kernel machines.

Despite the framing, that most reviewers found interesting, the execution left much to be desired among the reviewers. Among the many shared concerns are: the lack of clarity about the claimed contributions, it relation to an already sizeable literature that seeks to understand Zero (or Few) shot learning from a kernel / non-parametric perspective, and a limited experimental evaluation. After reading the paper and the reviews, I agree with the reviewers in that this paper, in its current form, is not ready for publication. In particular, it is not clear what the actual contribution is. Section 3 is a summary of well-known results about Kernel machines, Section 4 is mostly about sampling from a Kernel-based generative model, and, most critically, all of this is not even discussed in the context of LLMs, so the purported connection to zero-short learninng in LLMs is superficial.

Overall, I think this paper is clearly lacking is several aspects and therefore recommend rejection.

**Justification For Why Not Higher Score:**

The paper has important critical weaknesses that, without addressing, make it inadequate for acceptance.

**Justification For Why Not Lower Score:**

N/A

---

### Decision · Program_Chairs · 2024-01-16

Reject